# A Phenome-Wide Association Study of genes associated with COVID-19 severity reveals shared genetics with complex diseases in the Million Veteran Program

Anurag Verma[1,2,3☯]*, Noah L. Tsao[1,4☯], Lauren O. Thomann[5], Yuk-Lam Ho[6], Sudha K. Iyengar[7,8], Shiuh-Wen Luoh[9,10], Rotonya Carr[1,3,11], Dana C. Crawford[8,12,13], Jimmy T. Efird[14], Jennifer E. Huffman[5], Adriana Hung[15], Kerry L. Ivey[6,16,17], Michael G. Levin[4], Julie Lynch[18], Pradeep Natarajan[5,19,20], Saiju Pyarajan[5,21], Alexander G. Bick[5,22], Lauren Costa[6], Giulio Genovese[20,23,24], Richard Hauger[25], Ravi Madduri[26,27], Gita A. Pathak[28,29], Renato Polimanti[28,29], Benjamin Voight[1,2,30,31], Marijana Vujkovic[1,3], Seyedeh Maryam Zekavat[5,32,33], Hongyu Zhao[28,33,34,35], Marylyn D. Ritchie[2], VA Million Veteran Program COVID-19 Science Initiative, Kyong-Mi Chang[1,3], Kelly Cho[6], Juan P. Casas[6,36], Philip S. Tsao[37,38], J. Michael Gaziano[6,36], Christopher O'Donnell[5,21,36], Scott M. Damrauer[1,2,4‡], Katherine P. Liao[5,21,36‡]*

1 Corporal Michael Crescenz VA Medical Center, Philadelphia, Pennsylvania, United States of America, 2 Department of Genetics, Perelman School of Medicine, University of Pennsylvania, Philadelphia, Pennsylvania, United States of America, 3 Department of Medicine, Perelman School of Medicine, University of Pennsylvania, Philadelphia, Pennsylvania, United States of America, 4 Department of Surgery, Perelman School of Medicine, University of Pennsylvania, Philadelphia, Pennsylvania, United States of America, 5 VA Boston Healthcare System, Boston, Massachusetts, United States of America, 6 Massachusetts Veterans Epidemiology Research and Information Center (MAVERIC), VA Boston Healthcare System, Boston, Massachusetts, United States of America, 7 Louis Stokes Cleveland VA Medical Center, Cleveland, Ohio, United States of America, 8 Department of Population and Quantitative Health Sciences, Case Western Reserve University, Cleveland, Ohio, United States of America, 9 VA Portland Health Care System, Portland, Oregon, United States of America, 10 Division of Hematology and Medical Oncology, Knight Cancer Institute, Oregon Health and Science University, Portland, Oregon, United States of America, 11 University of Washington, Division of Gastroenterology, Seattle, Washington, United States of America, 12 Cleveland Institute for Computational Biology, Case Western Reserve University, Cleveland, Ohio, United States of America, 13 Department of Genetics and Genome Sciences, Case Western Reserve University, Cleveland, Ohio, United States of America, 14 Cooperative Studies Program Epidemiology Center, Health Services Research and Development, DVAHCS (Duke University Affiliate), Durham, North Carolina, United States of America, 15 Tennessee Valley Healthcare System (Nashville VA) & Vanderbilt University, Nashville, Tennessee, United States of America, 16 South Australian Health and Medical Research Institute, Infection and Immunity Theme, Adelaide, South Australia, Australia, 17 Harvard T.H. Chan School of Public Health, Department of Nutrition, Cambridge, Massachusetts, United States of America, 18 VA Informatics and Computing Infrastructure, VA Salt Lake City Health Care System, Salt Lake City, Utah, United States of America, 19 Cardiovascular Research Center, Massachusetts General Hospital, Boston, Massachusetts, United States of America, 20 Program in Medical and Population Genetics and the Cardiovascular Disease Initiative, Broad Institute of Harvard & MIT, Cambridge, Massachusetts, United States of America, 21 Harvard Medical School, Boston, Massachusetts, United States of America, 22 Department of Medicine, Vanderbilt University, Nashville, Tennessee, United States of America, 23 Stanley Center for Psychiatric Research, Broad Institute of MIT and Harvard, Cambridge, Massachusetts, United States of America, 24 Department of Genetics, Harvard Medical School, Boston, Massachusetts, United States of America, 25 Department of Psychiatry, University of California, San Diego, La Jolla, California; Center of Excellence for Stress and Mental Health, VA San Diego Healthcare System, San Diego, California, United States of America, 26 University of Chicago Consortium for Advanced Science and Engineering, The University of Chicago, Chicago, Illinois, United States of America, 27 Data Science and Learning Division, Argonne National Laboratory, Lemont, Illinois, United States of America, 28 VA Connecticut Healthcare System, West Haven, Connecticut, United States of America, 29 Department of Psychiatry, Yale School of Medicine, New Haven, Connecticut, United States of America, 30 Department of Systems Pharmacology and Translational Therapeutics, Perelman School of Medicine, University of Pennsylvania, Philadelphia, Pennsylvania, United

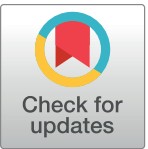

**Data Availability Statement:** Full summary statistics of the results presented in the study are made available. Individual level dataset underlying

this study cannot be shared outside the VA, except as required under the Freedom of Information Act (FOIA), per VA policy. However, upon request through the formal mechanisms in place and pending approval from the VHA Office of Research Oversight (ORO), a de-identified, anonymized dataset underlying this study can be created and shared. Upon request through the formal mechanisms provided by the VHA ORO, we would be able to provide sufficiently detailed variable names and definitions to allow replication of our work. Any requests for data access should be directed to the VHA ORO (OROCROW@va.gov), and should reference the following project and analysis: MVP035: A Phenome-Wide Association Study of genes associated with COVID-19 severity reveals shared genetics with complex diseases in the Million Veteran Program.

**Funding:** This research is based on data from the Million Veteran Program, Office of Research and Development, Veterans Health Administration, and was supported by award MVP035. S.M.D. is supported by US Department of Veterans Affairs (IK2-CX001780). R.C. is supported by NIH grants R01 AA026302 and P30 DK0503060. K.P.L. is supported by NIH P30 AR072577, and the Harold and Duval Bowen Fund. The funders had no role in study design, data collection and analysis, decision to publish, or preparation of the manuscript.

**Competing interests:** I have read the journal's policy and the authors of this manuscript have the following competing interests: RC has received research support from Intercept Pharmaceuticals, Inc and Merck & Co. SMD receives research support from RenalytixAI and personal consulting fees from Calico Labs, outside the scope of the current research. MDR is on the scientific advisory board for Goldfinch Bio and Cipherome. CO'D is an employee of Novartis Institute for Biomedical Research. PN reports grant support from Amgen, Apple, AstraZeneca, Boston Scientific, and Novartis, personal fees from Apple, AstraZeneca, Blackstone Life Sciences, Genentech, and Novartis, and spousal employment at Vertex, all unrelated to the present work.

States of America, **31** Institute for Translational Medicine and Therapeutics, Perelman School of Medicine, University of Pennsylvania, Philadelphia, Pennsylvania, United States of America, **32** Broad Institute of MIT & Harvard, Cambridge, Massachusetts, United States of America, **33** Yale School of Medicine New Haven, New Haven, Connecticut, United States of America, **34** Computational Biology and Bioinformatics Program, Yale University, New Haven, Connecticut, United States of America, **35** Department of Biostatistics, Yale School of Public Health, New Haven, Connecticut, United States of America, **36** Brigham and Women's Hospital, Boston, Massachusetts, United States of America, **37** VA Palo Alto Health Care System, Palo Alto, California, United States of America, **38** Department of Medicine (Cardiovascular Medicine), Stanford University School of Medicine, Stanford, California, United States of America

☯ These authors contributed equally to this work.
‡ These authors jointly supervised on this work.
* anurag.verma@pennmedicine.upenn.edu (AV); kliao@bwh.harvard.edu (KPL)

## Abstract

The study aims to determine the shared genetic architecture between COVID-19 severity with existing medical conditions using electronic health record (EHR) data. We conducted a Phenome-Wide Association Study (PheWAS) of genetic variants associated with critical illness (n = 35) or hospitalization (n = 42) due to severe COVID-19 using genome-wide association summary data from the Host Genetics Initiative. PheWAS analysis was performed using genotype-phenotype data from the Veterans Affairs Million Veteran Program (MVP). Phenotypes were defined by International Classification of Diseases (ICD) codes mapped to clinically relevant groups using published PheWAS methods. Among 658,582 Veterans, variants associated with severe COVID-19 were tested for association across 1,559 phenotypes. Variants at the *ABO* locus (rs495828, rs505922) associated with the largest number of phenotypes ($n_{rs495828} = 53$ and $n_{rs505922} = 59$); strongest association with venous embolism, odds ratio ($OR_{rs495828}$ 1.33 (p = 1.32 x $10^{-199}$), and thrombosis $OR_{rs505922}$ 1.33, p = 2.2 x$10^{-265}$. Among 67 respiratory conditions tested, 11 had significant associations including *MUC5B* locus (rs35705950) with increased risk of idiopathic fibrosing alveolitis OR 2.83, p = 4.12 × $10^{-191}$; *CRHR1 (*rs61667602) associated with reduced risk of pulmonary fibrosis, OR 0.84, p = 2.26× $10^{-12}$. The *TYK2* locus (rs11085727) associated with reduced risk for autoimmune conditions, e.g., psoriasis OR 0.88, p = 6.48 x$10^{-23}$, lupus OR 0.84, p = 3.97 x $10^{-06}$. PheWAS stratified by ancestry demonstrated differences in genotype-phenotype associations. *LMNA* (rs581342) associated with neutropenia OR 1.29 p = 4.1 x $10^{-13}$ among Veterans of African and Hispanic ancestry but not European. Overall, we observed a shared genetic architecture between COVID-19 severity and conditions related to underlying risk factors for severe and poor COVID-19 outcomes. Differing associations between genotype-phenotype across ancestries may inform heterogenous outcomes observed with COVID-19. Divergent associations between risk for severe COVID-19 with autoimmune inflammatory conditions both respiratory and non-respiratory highlights the shared pathways and fine balance of immune host response and autoimmunity and caution required when considering treatment targets.

## Author summary

Large population based genomic studies have discovered genetic variations associated with severe manifestations of Coronarvirus Disease 2019 (COVID-19). In this study, we

screened for other human conditions that share associations with these same variants. Understanding shared genetic variants in known conditions, where the pathophysiology is better understood, can further inform the pathways by which SARS-CoV2, the virus that causes COVID-19, impacts multiple organ systems. While genetic variants associated with severe COVID-19 were also associated with known risk factors and poor outcomes related to COVID-19 such as deep venous thrombosis, a large subset of these variants were also associated with reduced risk of conditions largely comprised of immune-mediated diseases, e.g., psoriasis, lupus, rheumatoid arthritis. With regards to the latter, the shared genetic architecture between COVID-19 and immune-mediated conditions suggests that pathways controlling both immune tolerance and immunodeficiency are important for COVID-19 severity, with implications when considering targeting these pathways for treatment.

## Introduction

Coronavirus disease 2019 (COVID-19) first identified in December of 2019[1], became a global pandemic by March 2020. As of September 2021, COVID-19, transmitted by the severe acute respiratory syndrome coronavirus 2 (SARS-CoV-2) virus, has resulted in the loss of over 5.4 million lives worldwide [2]. Identifying host genetic variants associated with severe clinical manifestations from COVID-19 can identify key pathways important in the pathogenesis of this condition. International efforts such as the COVID-19 Host Genetics Initiative (HGI)[3] have meta-analyzed genome-wide association study (GWAS) summary statistics at regular intervals to identify novel genetic associations with COVID-19 severity. Thus far, ten independent variants associated with COVID-19 severity at genome-wide significance have been identified, most notably at the *ABO* locus [4]. These GWASs have also identified variations in genes involving inflammatory cytokines and interferon signaling pathways such as *IFNAR2*, *TYK2*, and *DPP9* [4].

The unprecedented availability of genome-wide data for COVID-19 provides an opportunity to study clinical conditions that share genetic risk factors for COVID-19 severity. Examining known conditions, each with a body of knowledge regarding important pathways and targets, may in turn improve our understanding of pathways relevant for COVID-19 severity and inform the development of novel treatments against this pathogen. The Phenome-Wide Association Study (PheWAS) is an approach for simultaneously testing genetic variants' association with a wide spectrum of conditions and phenotypes [5]. The Veteran's Affairs (VA) Million Veteran Program (MVP) has generated genotypic data on over 650,000 participants linked with electronic health record (EHR) data containing rich phenotypic data, enables large-scale PheWAS. Moreover, MVP has the highest racial and ethnic diversity of the major biobanks worldwide affording an opportunity to compare whether associations are similar across ancestries [6].

The objective of this study was to use existing clinical EHR data to identify conditions that share genetic variants with COVID-19 severity using the disease-agnostic PheWAS approach. Since COVID-19 is a new condition, identifying existing conditions which share genetic susceptibility may allow us to leverage existing knowledge from these known conditions to provide context regarding important pathways for COVID-19 severity, as well as how pathways may differ across subpopulations.

## Methods

### Ethics statement

The Million Veteran Program received ethical and study protocol approval from the VA Central Institutional Review Board (IRB) in accordance with the principles outlined in the Declaration of Helsinki. All individuals in the study provided written informed consent as part of the MVP.

### Data sources

The VA MVP is a national cohort launched in 2011 designed to study the contributions of genetics, lifestyle, and military exposures to health and disease among US Veterans [6].

Blood biospecimens were collected for DNA isolation and genotyping, and the biorepository was linked with the VA EHR, which includes diagnosis codes (International Classification of Diseases ninth revision [ICD-9] and tenth revision [ICD-10]) for all Veterans followed in the healthcare system up to September 2019. The single nucleotide polymorphism (SNP) data in the MVP cohort was generated using a custom Thermo Fisher Axiom genotyping platform called MVP 1.0. The quality control steps and genotyping imputation using 1000 Genomes cosmopolitan reference panel on the MVP cohort has been described previously [7].

**Genetic variant selection.** An overview of the analytic workflow is outlined in Fig 1. Variants were derived from the COVID-19 HGI GWAS meta-analysis release v6 [3]. In this study, we analyzed the following HGI GWAS summary statistics: 1) hospitalized and critically ill COVID-19 vs. population controls denoted as "A2" in HGI, and referred to as "critical COVID" in this study, and 2) hospitalized because of COVID-19 vs. population controls, denoted as "B2" in HGI, referred to as "hospitalized COVID" in this study [3]. For each GWAS, variants with a Benjamini-Hochberg false discovery rate (FDR) corrected p-value < 0.01 were selected as candidate lead SNPs (3,502 associated with critical COVID, and 4,336 associated with hospitalized COVID). Variants with $r^2 <0.1$ were clustered within a 250 kb region according to 1000 Genomes phase 3 trans-ancestry reference panel [8]. Then, the variant with the smallest p-value in the region was selected as lead variant, resulting in 45 independent variants associated with critical COVID and 42 variants associated with hospitalized COVID summary statistics. The lead variants from each set of GWAS summary statistics are available in S1 Table. We used the nearest gene approach to prioritize the potential causal genes. A gene with the smallest genomic distance to a lead variant was selected.

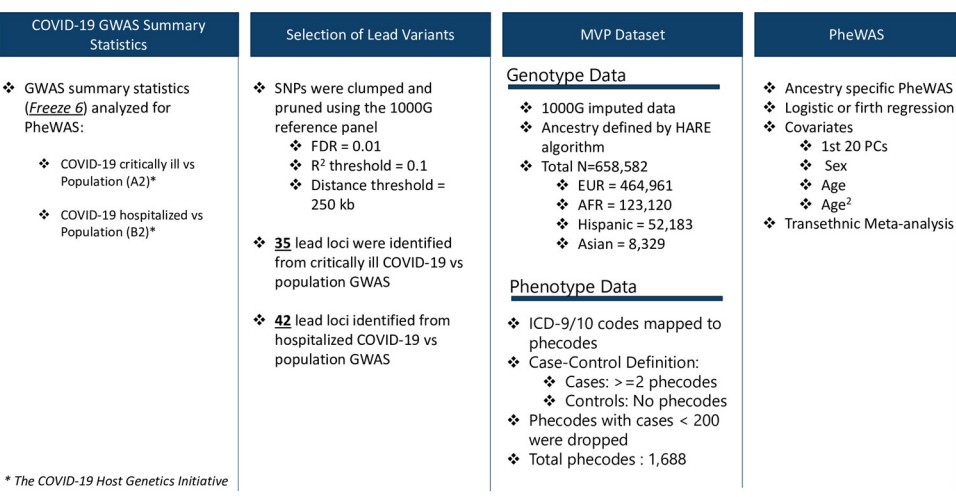

**Fig 1. Overview of variant selection and PheWAS analysis design.**

**Classification of race/ethnicity in MVP.** Race and ethnicity was determined using the harmonized ancestry and race/ethnicity (HARE) method for four major groups [9] corresponding to: (1) African ancestry, non-Hispanic Black (AFR), (2) Asian ancestry, non-Hispanic Asian (ASN), (3) European ancestry, non-Hispanic White (EUR), and (4) Hispanic ancestry (HIS). Individuals without a HARE classification were most likely be from ancestries with insufficient numbers to train the HARE algorithm including Native American, Alaska Native, and Pacific Islanders. Briefly, the HARE method was an algorithm developed to assign each subject to one of four groups using both self-reported race/ethnicity as the input to train a machine learning algorithm using ancestry informative genetic markers. This approach provides a classification for each individual leveraging information from genomic markers if self-reported race/ethnicity was missing, enabling analyses that can stratify by the four major groups. When comparing HARE vs self-reported race and ethnicity, the statistical error rate was estimated at 0.05–0.46%.

**Outcomes.** Clinical data prior to the onset of the COVID-19 pandemic were used to reduce potential confounding bias from SARS-CoV-2 infection on existing conditions. Phenotypes were defined by phecodes from prior studies [5,10]. Each phecode represents ICD codes grouped into clinically relevant phenotypes for clinical studies. For example, the phecode "deep venous thrombosis" includes "venous embolism of deep vessels of the distal lower extremities," and "deep venous thrombosis of the proximal lower extremity," both of which have distinct ICD codes. Using this approach, all ICD codes for all Veterans in MVP were extracted and each assigned a phenotype defined by a phecode. ICD-9 and ICD-10 codes were mapped to 1,876 phecodes, as previously described [5,10].

For each phecode, participants with $\geq 2$ phecode-mapped ICD-9 or ICD-10 codes were defined as cases, whereas those with no instance of a phecode-mapped ICD-9 or ICD-10 code were defined as controls. Based on our previous simulation studies of ICD EHR data, populations where the phecode comprises < 200 cases were more likely to result in spurious results [11], and we thus applied this threshold in four ancestry groups: AFR, ASN, HIS, and EUR. In total, we analyzed 1,617 (EUR), 1304 (AFR), 993 (HIS), 294 (ASN) phecodes from the MVP cohort.

**Phenome-wide association studies.** The primary PheWAS analysis used SNPs identified from the HGI GWAS of critical and hospitalized COVID, and tested association of these SNPs with phenotypes extracted from the EHR using data prior to the COVID-19 pandemic. Logistic regression using PLINK2 to examine the SNP association with phecodes and firth regression was applied when logistic regression model failed to converge. Regression models were adjusted for sex, age (at enrollment), age squared, and the first 20 principal components.

Ancestry-specific PheWAS was first performed in these four groups, and summary data were meta-analyzed using an inverse-variance weighted fixed-effects model implemented in the PheWAS R package [10]. We assessed heterogeneity using $I^2$ and excluded any results with excess heterogeneity ($I^2 > 40\%$).

To address multiple testing, an association between SNP and phecode with FDR p < 0.01 was considered significant. Thus, the threshold for significance was $p < 6.07 \times 10^{-05}$ for critical COVID lead variants, and $p < 4.13 \times 10^{-05}$ for hospitalized COVID lead variants. In the main manuscript we highlight PheWAS significant associations using FDR < 0.01 and an effect size associated with increased or reduced risk for a condition by 10%, with complete PheWAS results provided in S2 and S3 Tables.

## Results

We studied 658,582 MVP participants, with mean age 68 years, 90% male, with 30% participants from non-European ancestry (Table 1). The PheWAS was performed on 35 genetic

Table 1. Patient characteristics of Million Veteran Program participants.

| Characteristics | Million Veteran Program |
| --- | --- |
| | Number (%) |
| Total Patients | 658,582 |
| Male | 592,516 (90) |
| Ancestry | |
| European | 464,961 (70) |
| African | 123,120 (19) |
| Hispanic | 52,183 (8) |
| Asian | 8,329 (1) |
| Other | 9,989 (2) |
| Comorbidities | |
| Obesity (phecode = 278) | 283,197 (43) |
| Hypertension (phecode = 401.1) | 451,998 (69) |
| Type 2 Diabetes (phecode = 250.2) | 227,575 (34) |
| Coronary Artery Disease (phecode = 411.4) | 152,136 (23) |
| Chronic Kidney Disease (phecode = 585.2) | 10,046 (15) |

variants associated with critical COVID-19, and 42 genetic variants (S1 Table) associated with hospitalized COVID, across 1,559 phenotypes.

From the trans-ancestry meta-analysis, we identified 151 phenotypes significantly associated with critical COVID GWAS-identified variants, and 156 associations with hospitalized COVID GWAS-identified lead variants (FDR, p<0.01). Among these lead variants with significant PheWAS associations, 10 SNPs were associated with reduced risk of critical and hospitalized COVID-19 in HGI. Six variants were common to both critical and hospitalized COVID and had significant PheWAS associations, namely, variations nearest to the genes *ABO* (rs495828 and rs505922), *DPP9* (rs2277732), *MUC5B* (rs35705950), *TYK2* (rs11085727), and *CCHCR1* (rs9501257) (S2 and S3 Tables).

## Association of ABO loci with known risk factors and outcomes related to COVID-19 severity

In the trans-ancestry meta-analysis, the phenotype with the strongest association with variants near *ABO* locus (rs495828 and rs505922) was "hypercoagulable state" ($OR_{critical\_PheWAS}$ = 1.48 [1.42–1.54], $P_{critical\_PheWAS}$ = $1.84 \times 10^{-40}$; $OR_{hospitalized\_PheWAS}$ = 1.51 [1.46–1.56], $P_{hospitalized\_PheWAS}$ = $2.11 \times 10^{-55}$, Fig 2). The *ABO* loci had the largest number of significant PheWAS association findings, accounting for 35% (53/151) of significant phenotype associations in the critical COVID PheWAS, and 37% (59/156) in the hospitalized COVID PheWAS. The phenotypes with the most significant associations and largest effect size were related to hypercoagulable states and coagulopathies. As expected, conditions not related to coagulopathy associated with the *ABO* locus, included type 2 diabetes and ischemic heart disease, have been reported as risk factors for or are complications associated with COVID-19 severity and mortality [1,4,12,13] (Fig 2 and S2 and S3 Tables).

## Associations between variants associated with COVID-19 severity and respiratory conditions and infections

Among 68 respiratory conditions, only 11 diseases had significant associations (FDR < 0.01) shared with genetic variants associated with severe COVID-19. The most significant association was observed between rs35705950 (*MUC5B*) and idiopathic fibrosing alveolitis (OR = 2.83 [2.76–2.90]; $P$ = $4.12 \times 10^{-191}$), also known as idiopathic pulmonary fibrosis (IPF).

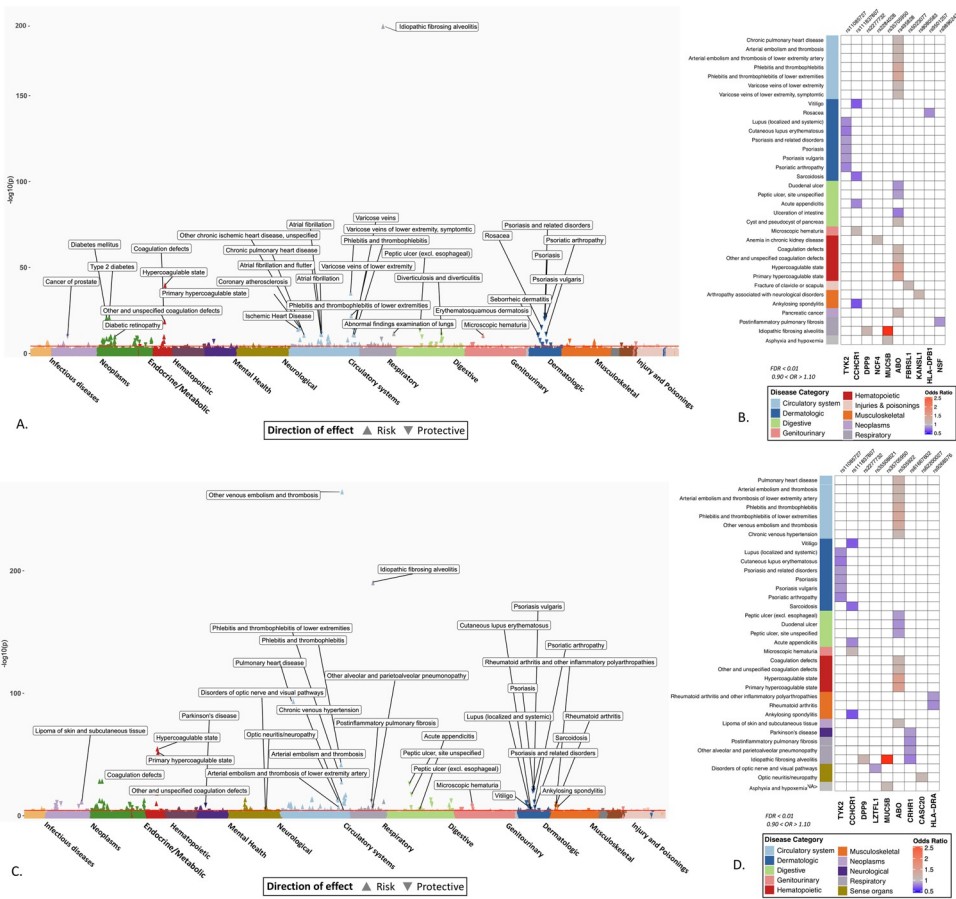

**Fig 2. PheWAS results of candidate SNPs from GWAS of critically ill and hospitalized COVID-19.** Significant associations between 48 SNPs from critical ill COVID GWAS (A) and 39 SNPs from hospitalized COVID (C) and EHR derived phenotypes in the Million Veteran Program. The phenotypes are represented on the x-axis and ordered by broader disease categories. The red line denotes the significance threshold using false discovery rate of 1% using the Benjamini-Hochberg procedure. The description of phenotypes is highlighted for the associations with FDR < 0.1 and odds ratio < 0.90 or odds ratio > 1.10. (B) and (D) A heatmap plot of SNPs with at least one significant association (FDR < 0.1). The direction of effect disease risk is represented by odds ratio. A red color indicates increased risk and blue color indicated reduced risk. The results with odds ratio < 0.90 or odds ratio > 1.10 are shown.

Similarly, rs2277732 near *DPP9* was associated with IPF (OR = 1.16 [1.09–1.22]; $P = 5.84 \times 10^{-06}$), both association between *MUC5B*, *DPP9* variants and IPF has been reported in previous studies [14]. However, the association of genetic variants with other respiratory conditions may represent novel findings: the association of intronic variant rs61667602 in *CRHR1* with reduced risk of post inflammatory pulmonary fibrosis (OR = 0.84 [0.80–0.89]; $P = 2.26 \times 10^{-12}$), "alveolar and parietoalveolar pneumonopathy" (OR = 0.80 [0.72–0.88]; $P = 1.58 \times 10^{-08}$) and IPF (OR = 0.87 [0.82–0.92], $P = 7.5 \times 10^{-07}$). We did not detect associations between any of the variants and other respiratory conditions which are known risk factors for COVID-19 such as chronic obstructive pulmonary disease (COPD, S2 and S3 Tables).

## Associations between variants associated with COVID-19 severity and reduced risk for certain phenotypes

The rs11085727-T allele of *TYK2*, a lead variant from the both critically ill and hospitalized COVID GWAS was associated with a reduced risk for psoriasis (OR = 0.88 [0.86–0.91],

**Table 2. Phenotypes sharing association with variants also associated with severe COVID-19 infection, with reduced odds of disease listed in order of p-value*.**

| Phenotype | OR (95% CI) | p-value | Gene | SNP | COVID-severity |
|---|---|---|---|---|---|
| Psoriasis | 0.89 [0.86–0.91] | 6.48E-23 | *TYK2* | rs11085727 | Both |
| Rosacea | 0.84 [0.8–0.89] | 7.54E-16 | *HLA-DPB1* | rs9501257 | Critical |
| Psoriatic arthropathy | 0.82 [0.77–0.88] | 6.97E-12 | *TYK2* | rs11085727 | Both |
| Post-inflammatory pulmonary fibrosis | 0.87 [0.83–0.92] | 4.54E-09 | *NSF* | rs9896243 | Critical |
| Vitiligo | 0.69 [0.56–0.82] | 3.03E-08 | *CCHCR1* | rs111837807 | Both |
| Sarcoidosis | 0.74 [0.62–0.85] | 1.80E-07 | *CCHCR1* | rs111837807 | Both |
| Lupus (localized and systemic) | 0.84 [0.77–0.91] | 3.97E-06 | *TYK2* | rs11085727 | Both |
| Cutaneous lupus erythematosus | 0.79 [0.68–0.89] | 6.21E-06 | *TYK2* | rs11085727 | Both |
| Post-inflammatory pulmonary fibrosis | 0.85 [0.8–0.9] | 2.26E-12 | *CRHR1* | rs61667602 | Hospitalized |
| Rheumatoid arthritis | 0.84 [0.79–0.9] | 4.20E-10 | *HLA-DRA* | rs9268576 | Hospitalized |
| Idiopathic fibrosing alveolitis | 0.81 [0.73–0.88] | 1.58E-08 | *CRHR1* | rs61667602 | Hospitalized |
| Rheumatoid arthritis and other inflammatory polyarthropathies | 0.88 [0.84–0.93] | 6.34E-08 | *HLA-DRA* | rs9268576 | Hospitalized |
| Other alveolar and parietoalveolar pneumonopathy | 0.88 [0.83–0.93] | 7.50E-07 | *CRHR1* | rs61667602 | Hospitalized |

*OR<0.9 and P<$10^{-5}$ shown in table, full results in supplementary; if multiple related conditions, e.g. psoriasis, psoriasis vulgaris, psoriasis and related disorders, description with lowest p-value selected shown in table.

$P = 6.48 \times 10^{-23}$), psoriatic arthropathy (OR = 0.82 [0.76–0.87], $P = 6.97 \times 10^{-12}$), and lupus (OR = 0.84 [0.76–0.91], $P = 63.97 \times 10^{-06}$). This *TYK2* signal has been previously reported to be associated with reduced risk of psoriasis, psoriatic arthropathy, type 1 diabetes, systemic lupus erythematosus and RA as well as other autoimmune inflammatory conditions (Table 2) [15,16].

## Ancestry specific PheWAS provide insights into differential disease risks

The PheWAS analyses performed across four major ancestry groups in MVP observed similar findings as the overall meta-analysis with few associations unique to a specific ancestry (Fig 3 and S8 Table). SNP rs581342 (*LMNA*), associated with severe COVID-19, was a highly prevalent variant among subjects with AFR ancestry (MAF = 0.53) and was associated with neutropenia ($OR_{AFR} = 1.29$ [1.21–1.39] $P_{AFR} = 4.09 \times 10^{-13}$); this association was also observed in HIS ancestry ($OR_{HIS} = 1.65$ [1.32–2.06], $P_{HIS} = 8.84 \times 10^{-06}$) but was not in the larger EUR ancestry (S8 Table). To follow-up on this association, we extracted data on laboratory values for white blood cell (WBC) count and neutrophil fraction on all subjects. *LMNA* was associated with lower WBC in AFR, EUR, and HIS ancestries. However, *LMNA* was associated with a lower median neutrophil fraction only among Veterans of AFR ancestry (beta = -1.84 [-1.94, -1.75], $P_{AFR} = 1 \times 10^{-300}$) and HIS ancestry (beta = -0.67, $P_{HIS} = 7.2 \times 10^{-13}$) but not among Veterans of EUR ancestry (beta = -0.09, $P_{EUR} = 0.005$). Among individuals of AFR ancestry, each allele was associated with a 1.84% lower neutrophil fraction, where among individuals of HIS and EUR ancestry, each allele was associated with 0.67% and 0.04% reduction, respectively (S9 Table).

Similarly, associations between rs9268576 (*HL-DRA*) and thyrotoxicosis was only observed in individuals of AFR ancestry. The EUR ancestry specific PheWAS identified 39 significant associations which were not observed in other ancestry groups. One such association was between *MUC5B* variant and phecode for "dependence on respirator [ventilator] or supplemental oxygen" ($OR_{EUR} = 1.16$ [1.11–1.12], $P_{EUR} = 1.72 \times 10^{-10}$) among individuals of EUR ancestry was not significant in other ancestry groups (S8 Table). It is important to note that the conditions with significant association among individuals of EUR ancestry had similar

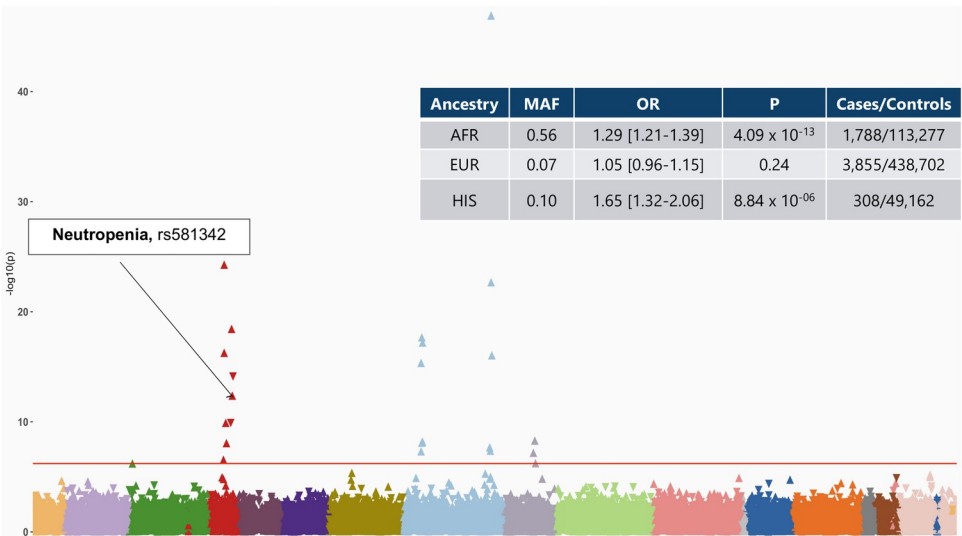

| Ancestry | MAF | OR | P | Cases/Controls |
|---|---|---|---|---|
| AFR | 0.56 | 1.29 [1.21-1.39] | $4.09 \times 10^{-13}$ | 1,788/113,277 |
| EUR | 0.07 | 1.05 [0.96-1.15] | 0.24 | 3,855/438,702 |
| HIS | 0.10 | 1.65 [1.32-2.06] | $8.84 \times 10^{-06}$ | 308/49,162 |

**Fig 3. PheWAS results of candidate SNPs from GWAS of Hospitalized COVID-19 in individuals of AFR ancestry.** The plot highlights the association between rs581342 SNP and neutropenia, which was only observed in the AFR ancestry. The phenotypes are represented on the x-axis and ordered by broader disease categories. The red line denotes the significance threshold using false discovery rate of 1% using the Benjamini-Hochberg procedure. The table on the top right of the plot shows the association results between rs581342 and neutropenia in other ancestries. The association was not tested among participants of ASN ancestry due to low case numbers.

prevalence among other ancestries. However, since there were overall fewer subjects in non-EUR ancestry groups, this likely resulted in lower statistical power to detect associations. All ancestry specific PheWAS results are available in supplementary tables (S4, S5, S6, and S7 Tables, and S1 and S2 Figs).

## Association with variation at sex chromosome

In the hospitalized COVID-19 GWAS, we identified rs4830964 as the only lead variant on chromosome X. The SNP is located near *ACE2* and was associated with "non-healing surgical wound" (OR = 0.92 [0.89–0.96], $P = 2.23 \times 10^{-05}$). Notably, the SNP had nominal association (p<0.05) with type 2 diabetes and diabetes related complications that are previously reported association with variation in *ACE2* (S3 Table). We did not observe any association with this variant in the ancestry specific PheWAS analysis.

## Discussion

In this large-scale PheWAS, we identified the shared genetic architecture between variants associated with severe COVID-19 and other complex conditions using data from MVP, one of the largest and most diverse biobanks in the world. Broadly, these risk alleles identified conditions associated with risk factors for severe COVID-19 manifestations such as type 2 diabetes and ischemic heart disease across all ancestries examined here. Notably, the strongest associations with the highest effect size were related to coagulopathies, specifically, hypercoagulable state including deep venous thrombosis and other thrombotic complications, also shared variants associated with severe COVID-19. In contrast, among respiratory conditions, only idiopathic pulmonary fibrosis and chronic alveolar lung disease shared genetic risk factors, with the notable absence of an association with COPD and other respiratory infections. When comparing findings across ancestry groups in MVP, we observed that a risk allele associated with severe COVID-19, *LMNA*, also shared an association with neutropenia among Veterans of

AFR and HIS ancestry. Finally, we observed that variants associated with severe COVID-19 had an opposite association, or reduced odds with autoimmune inflammatory conditions, such as psoriasis, psoriatic arthritis, RA, and inflammatory lung conditions.

A classic GWAS tests the association between millions of genetic variants with the presence or absence of one phenotype, e.g., GWAS of deep venous thrombosis. In the COVID-19 HGI GWAS, the "phenotype" was patients hospitalized for or critically ill from COVID-19. Clinically, this population includes a mixture of patients with a complex list of medical conditions at high risk for severe COVID complications and those who had actual complications from COVID-19. Thus, we would anticipate that many of the significant phenotypes would be associated with risk factors such as obesity and deep venous thrombosis. Additionally, our findings suggest that the PheWAS approach can be a useful tool to identify clinical factors related to emerging infectious diseases regarding severity or complications when genomic data are available.

The PheWAS results of SNPs in the *ABO* locus served as a positive control for this study. Genetic variations in *ABO* are an established risk factor for COVID-19 severity. Patients with blood group A have a higher risk of requiring mechanical ventilation and extended ICU stay compared with patients with blood group O [17]. These same variations at *ABO* had known associations with a spectrum of blood coagulation disorders identified in studies pre-dating COVID-19 [18–20]. The PheWAS of *ABO* variants identified associations with increased risk of deep vein thrombosis, pulmonary embolism, and other circulatory disorders, in line with prior studies, and recent studies among patients hospitalized with COVID-19 [21–25].

Among the respiratory conditions, only idiopathic pulmonary fibrosis (IPF) and chronic alveoli lung disease had shared associations with the variants near genes *MUC5B*, *CRHR1*, and *NSF*. Located in the enhancer region of the *MUC5B*, rs35705950, is a known risk factor for IPF, and a high mortality rate was observed among the COVID-19 patients with pre-existing IPF [26]. However, the *MUC5B* variant is associated with a reduced risk of severe COVID-19 (OR = 0.89), suggesting the risk allele's opposing effect for infection and pulmonary fibrosis. In a separate study of MVP participants tested for COVID-19, we identified a significant mediating effect of the *MUC5B* variant in reducing risk for pneumonia due to COVID-19 [27].

Several conditions shared genetic variants associated with severe COVID-19, however, the association was for reduced odds for these conditions. All except one, rosacea, have an autoimmune etiology. The existing literature can help explain some of the dual association between reduced risk of autoimmune conditions such as psoriasis and RA and increased risk of severe COVID-19 via *TYK2*. *TYK2*, a member of the Janus Kinase (*JAK*) family of genes, plays a key role in cytokine signal transduction and the inflammatory response [28], specifically in type 1 interferon signaling, part of the innate immune response blocking the spread of a virus from infected to uninfected cells. Partial loss of *TYK2* function is associated with reduced risk for several autoimmune disorders such as RA and psoriatic disease, conditions treated with immunosuppressive therapy [15,29–32]. Humans with complete *TYK2* loss of function have clinically significant immunodeficiency with increased susceptibility to mycobacterial and viral infections [28,33]. Thus, this observation for opposing associations of variants with COVID-19 and autoimmune conditions highlights the fine balance between host immune response and autoimmunity.

While non-white populations are disproportionately affected by COVID-19, the majority of studies still predominantly consist of individuals from EUR ancestry. The COVID-19 GWAS data from the HGI consists of participants from over 25 countries EUR (33% non-EUR samples), enabling identification of variants more prevalent in non-EUR populations. We used these data to perform a PheWAS on the linked genotype-phenotype data from MVP, the most racial and ethnically diverse biobank in the US. From this large-scale study across ancestries, we observed that a variant located in the *LMNA* gene locus was associated with a

diagnosis of neutropenia in AFR ancestry and HIS, but not EUR which would otherwise would have been well powered to detect an association. *LMNA* was associated with lower WBC counts across all ancestries, but its association with a lower fraction of neutrophils was observed in AFR and HIS only, but not EUR, in line with the overall association with diagnoses codes for neutropenia. ASN ancestry comprised the smallest ancestry group in MVP and was not tested due to low case numbers for neutropenia.

*LMNA* variants are associated with a broad spectrum of cardiomyopathies such as dilated cardiomyopathies, familial atrial fibrillation. However, the association with neutropenia has not been previously reported. Neutropenia refers to an abnormally low number of neutrophils in the blood, and predisposing to increased risk of infection. Epidemiology studies have shown that lower neutrophil counts are more common in individuals of AFR ancestry [34,35] and are hypothesized to be a result of selection and generally considered benign. To our knowledge benign neutropenia has not been previously reported among individuals of HIS ancestry [36]. Whether low neutrophil levels may clinically impact COVID-19 outcomes remains to be seen and warrants further study.

## Limitations

The PheWAS of risk alleles associated with severe COVID-19 did not observe an association between other chronic pulmonary conditions such as COPD, a risk factor for severe COVID manifestations [13,37,38]. This absence of association allows us to discuss a few limitations of the PheWAS approach. The PheWAS was designed to broadly screen for potentially clinically relevant associations between genes and thousands of phenotypes. The phenotypes are based on ICD codes, and the accuracy of these codes can vary across conditions. Misclassification of cases and controls would reduce power to detect associations. The clinical definition of COPD itself is an area of active discussion and thus could impact the already modest accuracy of COPD diagnostic codes, further limiting power to detect an association [39,40]. As well, the PheWAS has limited power to detect associations for uncommon conditions and may explain the absence of associations with another chronic pulmonary condition, CF which has a prevalence of 0.02% in this Veteran population. Alternatively, studies to date have yielded mixed results with regards to risk for severe COVID among patients with CF [41]. To enable a trans-ancestry study, we applied a conservative approach using 20 PCs to adjust all models, used in prior studies. One potential pitfall of this approach is that the models may be overadjusted and thus are more likely to miss few significant associations. Finally, COPD is a condition where cigarette smoking, an environmental risk factor, accounts for the majority of cases [42]. While genetics is an important aspect of COPD, the link between variants and COPD may be weaker compared to other conditions where genetic variants drive the phenotype, ABO blood type as an example. Thus, conditions such as COPD where environmental risk factors or where gene-environment interactions play a major role in risk, may be more difficult to identify in a standard PheWAS. Findings from this study suggest that variants associated with severe COVID-19 are also associated with reduced odds of having an autoimmune inflammatory condition. However, the results cannot provide information on the impact of actual SARS-CoV-2 infection in these individuals after diagnosis of an autoimmune disease.

## Conclusions

The PheWAS of genetic variants reported to associate with severe COVID-19 demonstrated shared genetic architecture between COVID-19 severity and known underlying risk factors for both severe COVID-19 and poor COVID-19 outcomes, rather than susceptibility to other viral infections. Overall, the associations observed were generally consistent across genetic

ancestries, with the exception of a stronger association with neutropenia among Veterans of AFR and HIS ancestry and not EUR. Notably, only few respiratory conditions had a shared genetic association with severe COVID-19. Among these, variants associated with a reduced risk for severe COVID-19 had an opposite association, with reduced risk for inflammatory and fibrotic pulmonary conditions. Similarly, other divergent associations were observed between severe COVID-19 and autoimmune inflammatory conditions, shedding light on the concept of the fine balance between immune tolerance and immunodeficiency. This balance will be important when considering therapeutic targets for COVID-19 therapies where pathways may control both inflammation and the viral host response.

## Supporting information

**S1 Table. List of lead variants from critical ill and hospitalized COVID GWAS included in the study.**
(XLSX)

**S2 Table. Meta-analysis summary statistics from PheWAS of 35 lead SNPs identified from critical ill COVID GWAS.**
(XLSX)

**S3 Table. Meta-analysis summary statistics from PheWAS of 42 lead SNPs identified from Hospitalized COVID GWAS.**
(XLSX)

**S4 Table. Summary statistics from EUR ancestry PheWAS of lead SNPs identified from critical ill and hospitalized COVID-19 GWAS.**
(XLSX)

**S5 Table. Summary statistics from AFR ancestry PheWAS of lead SNPs identified from critical ill and hospitalized COVID-19 GWAS.**
(XLSX)

**S6 Table. Summary statistics from HIS ancestry PheWAS of lead SNPs identified from critical ill and hospitalized COVID-19 GWAS.**
(XLSX)

**S7 Table. Summary statistics from ASN ancestry PheWAS of lead SNPs identified from critical ill and hospitalized COVID-19 GWAS.**
(XLSX)

**S8 Table. Ancestry specific comparison of PheWAS results.**
(XLSX)

**S9 Table. Ancestry specific comparison of association between rs581342 and median values of neutrophil fraction and white blood cell counts.**
(XLSX)

**S1 Fig.** The PheWAS results of 48 SNPs from critical ill COVID GWAS by each ancestry a) European ancestry, b) African ancestry, c) Hispanic ancestry, and d) Asian ancestry.
(TIF)

**S2 Fig.** The PheWAS results of 39 SNPs from hospitalized COVID GWAS by each ancestry a) European ancestry, b) African ancestry, c) Hispanic ancestry, and d) Asian ancestry.
(TIF)

**S1 Text. VA Million Veteran Program COVID-19 Science Initiative Membership & Acknowledgements.**
(DOCX)

## Acknowledgments

We are grateful to our Veterans for their contributions to MVP. Full acknowledgements for the VA Million Veteran Program COVID-19 Science Initiative can be found in the S1 Text. We would like to thank the Host Genetic Initiative for making their data publicly available (https://www.covid19hg.org/acknowledgements/). This publication does not represent the views of the Department of Veteran Affairs or the United States Government.

## Author Contributions

**Conceptualization:** Anurag Verma, Scott M. Damrauer, Katherine P. Liao.

**Data curation:** Anurag Verma, Noah L. Tsao, Yuk-Lam Ho, Jennifer E. Huffman.

**Formal analysis:** Anurag Verma, Noah L. Tsao.

**Funding acquisition:** Kyong-Mi Chang, Scott M. Damrauer, Katherine P. Liao.

**Methodology:** Anurag Verma.

**Project administration:** Lauren O. Thomann, Lauren Costa.

**Resources:** Lauren O. Thomann, Yuk-Lam Ho, Kelly Cho.

**Supervision:** Philip S. Tsao, J. Michael Gaziano, Christopher O'Donnell, Scott M. Damrauer, Katherine P. Liao.

**Visualization:** Anurag Verma.

**Writing – original draft:** Anurag Verma, Noah L. Tsao, Scott M. Damrauer, Katherine P. Liao.

**Writing – review & editing:** Anurag Verma, Noah L. Tsao, Sudha K. Iyengar, Shiuh-Wen Luoh, Rotonya Carr, Dana C. Crawford, Jimmy T. Efird, Jennifer E. Huffman, Adriana Hung, Kerry L. Ivey, Michael G. Levin, Julie Lynch, Pradeep Natarajan, Saiju Pyarajan, Alexander G. Bick, Giulio Genovese, Richard Hauger, Ravi Madduri, Gita A. Pathak, Renato Polimanti, Benjamin Voight, Marijana Vujkovic, Seyedeh Maryam Zekavat, Hongyu Zhao, Marylyn D. Ritchie, Kyong-Mi Chang, Kelly Cho, Juan P. Casas, Philip S. Tsao, J. Michael Gaziano, Christopher O'Donnell, Scott M. Damrauer, Katherine P. Liao.

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
