## [Decision Letter · Decision Letter 0]

2 Dec 2021

Dear Dr Verma,

Thank you very much for submitting your Research Article entitled 'A Phenome-Wide Association Study of genes associated with COVID-19 severity reveals shared genetics with complex diseases in the Million Veteran Program' to PLOS Genetics.

The manuscript was fully evaluated at the editorial level and by independent peer reviewers. The reviewers appreciated the attention to an important problem, but raised some substantial concerns about the current manuscript. Based on the reviews, we will not be able to accept this version of the manuscript, but we would be willing to review a much-revised version. We cannot, of course, promise publication at that time.

If you decide to revise the manuscript for further consideration at PLOS Genetics, please aim to resubmit within the next 60 days, unless it will take extra time to address the concerns of the reviewers, in which case we would appreciate an expected resubmission date by email to plosgenetics@plos.org.

[LINK]

We are sorry that we cannot be more positive about your manuscript at this stage. Please do not hesitate to contact us if you have any concerns or questions.

Yours sincerely,

Gregory S. Barsh

Editor-in-Chief

PLOS Genetics

Gregory Copenhaver

Editor-in-Chief

PLOS Genetics

Reviewer's Responses to Questions

**Comments to the Authors:**

Reviewer #1: I was involved in the review of this work when it was submitted to PLoS Medicine. I was invited to review the present manuscript as a revision to that submission. Overall, the paper is a substantial improvement over the previous version and addresses the majority of my earlier concerns. In particular, there is a much stronger justification for the choice of parameter values, and the removal of UKBB greatly improves the clarity of presentation. I only have one relatively minor concern remaining.

While it is common to adjust for the first k principal components (PCs) as a way to correct for broad-scale population structure, it may be good to show that the first 20 PCs do NOT align with the variables being modeled (associations with phecodes in this case). Otherwise, there is a concern that adjusting for PCs may lower the signal-to-noise ratio and artificially deflate the importance of one or more of the phecodes being investigated. The artificial deflation of importance may, for example, explain why the authors did not detect some of the known risk factors, such as COPD (Lines 268, 320).

Minor comments:

-eTable 1 -> S1 Table.

-The ancestry group acronyms EUR, AFR, etc. are used before they are defined.

-Line 237: “both severe and hospitalized COVID” -> “both critical and hospitalized COVID”

-Line 253: “have been reported as risk factors for or are complications associated with COVID-19 severity and mortality” -should cite the work being referenced in the sentence

-Figures 2A&C: Some of the labels are obscured by the overlap.

-Line 314: The acronym T2D (Type 2 Diabetes) has not been defined

-Line 332: The sentence “The clinical data used in this study pre-dates the emergence of COVID-19…” feels out of place and repeats what was stated in the Methods. Consider cutting.

Reviewer #2: This manuscript reports a carefully executed and clearly described phenome wide association study (PheWAS) of SNPs associated with critical and hospitalized COVID in over 650K ancestrally diverse participants in the Million Veteran Program (MVP). The Methods are in general well defined, the Results clearly presented, and the Discussion clear, though in places a bit verbose.

Given that its ethnic and ancestral diversity is a major strength of the MVP, it is surprising that the authors reported their findings by genetic ancestry groups alone and not by self-identified race/ethnic groups as well. While the two are highly correlated, they are not identical, and at minimum a description of the correlation between the two should be provided. Preferable would be to repeat the findings stratifying by self-identified groups and describing observed similarities and differences. MVP is one of the very few cohorts that can do this; this opportunity should not be missed. They should also describe how individuals who did not fit into one of the four HARE groups were analyzed.

The Methods do not describe how SNPs were assigned to genes (or vice versa). Typically 85-90% of GWAS associations lie in intergenic regions, some of which are known to play important regulatory roles in other genes. The only reference I could find was in line 238, simply, “…variations nearest to the genes….” This should be described in more detail, as should the search for important regulatory regions tagged by non-exonic SNPs. They should also describe how the SNPs in Table 2 and Figs 2 and 3 were selected as “representative” of these genes. Presumably they chose the SNP with the strongest association (by p-value?) but this does not seem to be stated anywhere.

The data table in Fig. 3 describe a stronger association of rs581342 and neutropenia in COVID hospitalized persons of Hispanic genetic ancestry than in those of African ancestry (OR 1.65, p 8.84 x 10-6 ) that exceeds the authors’ defined significance threshold of p < 4.13 × 10-05, yet it is never mentioned. While the stronger OR is a point estimate based on a small number of cases (318 HIS vs. 1,788 AFR) and its confidence interval likely overlaps with that in AFR, the significance level merits a discussion despite its not being consonant with the authors’ hypothesis of a relationship to the known benign neutropenia among African Americans. Or is there an error in the table?

The authors go into extensive discussions of PheWAS associations with idiopathic pulmonary fibrosis (lines 348-364), autoimmune diseases (lines 366-83) and neutropenia (lines 398-405) but provide almost no explanation of the observed lack of associations with pre-existing pulmonary disease aside from reiterating them in lines 319-20. The former three discussions could be cut back (particularly the seemingly tangential link to depression and anxiety in asthma in lines 360-61) and expand on reasons for the surprising lack of pulmonary disease associations or suggest research to clarify them.

Minor comments:

1. Line 360: There may be a word missing in “…variations in this gene have also shown associations enhanced improvement….”

2. Table 1: numbers of Asian, Other, and Chronic Kidney Disease seem to be incorrectly punctuated.

**Have all data underlying the figures and results presented in the manuscript been provided?**

Reviewer #1: Yes

Reviewer #2: **No: **Restrictions on access to VA data are beyond the authors' control but they do describe procedures for accessing the data.

PLOS authors have the option to publish the peer review history of their article (what does this mean?). If published, this will include your full peer review and any attached files.

Reviewer #1: **Yes: **Artem Sokolov

Reviewer #2: **Yes: **Teri Manolio

---

## [Decision Letter · Decision Letter 1]

20 Feb 2022

Dear Dr Verma,

We are pleased to inform you that your manuscript entitled "A Phenome-Wide Association Study of genes associated with COVID-19 severity reveals shared genetics with complex diseases in the Million Veteran Program" has been editorially accepted for publication in PLOS Genetics. Congratulations!

The revised manuscript was seen by both of the original reviewers and as you will see they are enthusiastic about moving forward (as are we).

Yours sincerely,

Gregory S. Barsh

Editor-in-Chief

PLOS Genetics

Gregory Copenhaver

Editor-in-Chief

PLOS Genetics

Comments from the reviewers (if applicable):

Reviewer's Responses to Questions

**Comments to the Authors:**

Reviewer #1: Thank you for another opportunity to consider this manuscript. I really appreciate the authors sharing their insight into how the 20 principal components were chosen. The revisions address all of my remaining concerns, and I am happy to recommend this work for publication.

Reviewer #2: The authors have addressed my concerns.

**Have all data underlying the figures and results presented in the manuscript been provided?**

Reviewer #1: Yes

Reviewer #2: Yes

PLOS authors have the option to publish the peer review history of their article (what does this mean?). If published, this will include your full peer review and any attached files.

Reviewer #1: **Yes: **Artem Sokolov

Reviewer #2: **Yes: **Teri A Manolio

**Data Deposition**

http://datadryad.org/submit?journalID=pgenetics&manu=PGENETICS-D-21-01344R1

**Press Queries**

---

## [Editor Report · Acceptance letter]

4 Apr 2022

PGENETICS-D-21-01344R1 

A Phenome-Wide Association Study of genes associated with COVID-19 severity reveals shared genetics with complex diseases in the Million Veteran Program 

Dear Dr Verma, 

We are pleased to inform you that your manuscript entitled "A Phenome-Wide Association Study of genes associated with COVID-19 severity reveals shared genetics with complex diseases in the Million Veteran Program" has been formally accepted for publication in PLOS Genetics! Your manuscript is now with our production department and you will be notified of the publication date in due course.

With kind regards,

Agnes Pap

PLOS Genetics

On behalf of:
